# Sensitivity—Bandwidth Optimization of PMUT with Acoustical Matching Using Finite Element Method

**DOI:** 10.3390/s22062307

**Published:** 2022-03-16

**Authors:** Le-Ming He, Wei-Jiang Xu, Yan Wang, Jia Zhou, Jun-Yan Ren

**Affiliations:** 1State Key Laboratory of ASIC and System, School of Microelectronics, Fudan University, Shanghai 201203, China; lmhe13@fudan.edu.cn (L.-M.H.); yan_wang19@fudan.edu.cn (Y.W.); jia2000@fudan.edu.cn (J.Z.); 2Université Polytechnique Hauts-de-France, CNRS, Université de Lille, Centrale Lille, UMR 8520-IEMN, DOAE, F-59313 Valenciennes, France; 3INSA Hauts-de-France, Le Mont Houy, F-59313 Valenciennes, France

**Keywords:** PMUTs, FEM, optimization, acoustic matching, round-trip, sensitivity, bandwidth

## Abstract

A new model in finite element method to study round-trip performance of piezoelectric micromachined ultrasonic transducers (pMUTs) is established. Most studies on the performance of pMUT are based only on the transmission sensibility, but the reception capacity is as much important as the transmission one, and is quite different from this latter. In this work, the round-trip sensitivity of pMUT is defined as the product of the frequency response of transmitted far field pressure to source voltage excitation and that of reception output to return wave pressure. Based on this sensitivity characteristic, firstly, a multi-parameter optimization for a cavity pMUT is performed using the sensitivity-bandwidth product parameter SBW as criterion. The radii of the electrode and the piezoelectric layer, the thicknesses of the piezoelectric layer and the vibration diaphragm are adjusted to maximize the performance. Secondly, an acoustic matching method is proposed and applied to pMUTs for the first time. As a result, the round-trip sensitivity can be evaluated and the pulse-echo response of wide-band excitation can be simulated, giving the most quantitative and intuitive feedback for pMUT design. The optimization enhances the sensitivity-bandwidth product by 52% when the top electrode and piezoelectric layer are both etched to 75% radius of the cavity beneath; the introduction of an acoustic matching layer shows significant bandwidth expansion in both the transmitting and receiving process.

## 1. Introduction

Ultrasound is widely applied in medical imaging, nondestructive testing, intelligent sensing and so on. To convert electrical energy into acoustic energy and vice-versa, an ultrasonic transducer is a must. As imaging applications become able to see tinier objects (tens of microns) [1], the transducer faces challenges of working at a higher frequency (above 10 MHz) to increase image resolution, while having high sensitivity and bandwidth, and sometimes even miniature size, for example, as demanded in inter-vascular ultrasound [2].

Piezoelectric micromachined ultrasonic transducers (pMUT) can be the ideal candidate for high frequency ultrasonic imaging. A pMUT uses Micro-Electro-Mechanical Systems (MEMS) process to achieve its structural patterning, film deposition, cavity etching, etc. [3]. The fabrication technology is highly compatible with the semiconductor process, thus the pMUTs can be easily integrated with the electrical excitation source (HV pulser) and analog front end (AFE) chip to minimize the system [4,5]. The MEMS process improves the cell uniformity of large-scale array pMUTs. Compared with traditional bulk piezoelectric transducers (based on thickness resonance mode) and capacitive micromachined ultrasonic transducer (cMUT), pMUT is more robust and has a high yield rate. However, the actual developed pMUT lacks sufficient sensitivity and bandwidth to perform high frequency ultrasonic imaging, hence to retrieve clear, accurate image of the target.

To resolve the above-mentioned shortcomings, new pMUT structures are being proposed for the purpose of increasing sensitivity and bandwidth [6,7]. However, before the fabrication of a new design, efficient modeling, simulation and parameter optimization are very important and necessary. The working principle of pMUT is based on the vibration of a thin diaphragm with its border clamped on a silicon substrate, driven by a piezoelectric film deposited on the diaphragm surface. In the most-used modeling approach—Timoshenko’s theory of plates—the deposited sandwich piezoelectric structure, together with passive supporting layer are regarded as a uniform circular thin plate (or disk) with fixed boundary. The harmonic vibration solutions obtained from the plate motion equation are flexural bending modes. When excited, the piezoelectric effect is equivalent to a moment force applied at the equivalent plate surface, actuating the plate vibration [8,9], and transmitting sound energy if the plate is loaded.

Timoshenko’s theory simulates the mechanical behavior of the plate in a 2D approximation, but cannot describe the pMUT’s stress-displacement in the plate thickness. Another approach is the derivative equivalent circuit models, including Mason’s model [10,11] that associates an electrical port to an acoustical one using an electro-acoustical transformer, and BDV (Butterworth-Van Dyke) model [12] that is a lumped-parameter equivalent circuit describing pMUT electro-mechanical behavior near its resonance. The above-cited models concisely explain the transduction behavior, but can only solve electromechanical coefficients or displacement on the thin film’s surface [13,14]. They over-simplify the pMUT structure and cannot fully cover its transmitting and receiving process, limiting hence their utilization in the developing and optimization of pMUT.

Recently, the modeling works shift to the use of the finite element method (FEM). The method is based on discretizing partial differential equations (PDEs) into a system of linear algebraic equations, which solve complicated multi-physics-coupled models while complying with the original constitutive laws and governing equations. By setting up a 2-D axisymmetric or 3-D model, one can calculate electro-mechanical-acoustical properties of pMUT accurately [15,16]. The studies on pMUT are making progress on proposing new structures and doing structure optimization. In the optimization, certain criteria should be first defined. As the key performance parameters of pMUT, the conversion efficiency, i.e., the transmission sensibility between source voltage and transmitted sound pressure and the bandwidth are usually the main criteria. However the pMUT is not a reciprocal system, which means that its transmission from excitation voltage to sound pressure is not the same as its reception from sound pressure to the voltage [17,18]. Therefore, a round-trip analysis of pMUTs, including both the transmission and reception sensibility, is necessary. Moreover, only the FEM provides the convenience for such modeling.

The most-used criteria in thickness mode transducers are round-trip sensitivity (in frequency domain) and pulse-echo waveform (in time domain), which are more comprehensive and straightforward. By reviewing the prior works, the available experiment results of round-trip sensitivity of pMUTs are summarized in Figure 1, and compared with cMUTs and thickness mode transducers. Apparently, the overall performance of MUTs is not as good as the thickness mode transducers, hence it is very necessary and urgent to improve the performance in both sensitivity and bandwidth.

In this work, we use the finite element method to study the round-trip performance of pMUTs and propose a multi-parameter optimization strategy. Starting with a desired working frequency (10 MHz), the geometry parameters, including the top electrode size and the piezoelectric film radius and thickness of the pMUT are optimized to obtain maximum round-trip gain-bandwidth product. A new design of pMUT with acoustic matching is introduced, and its thickness is optimized based on the same criterion. The study is mainly divided into three parts:

Modeling Work: The round-trip analysis has been established by solving transmitting and receiving models separately, thus realizing pulse-echo test simulation of pMUTs in full frequency band. Based on the frequency domain round-trip response, the pulse-echo waveform excited by wide-band signal in the time domain can be simulated. Such simulation gives the most quantitative and intuitive feed-back to designers, and is very helpful for innovation and optimization of pMUTs.

Optimization Work: Based on the above modeling method, a more significant and robust performance indicator has been proposed: i.e., the product of round-trip sensitivity and fractional bandwidth (SBW), taking into account not only the influence of the complex geometry on sensitivity and bandwidth of the pMUT, but also the whole wave process from source excitation to charged reception including coupling medium. Based on the SBW criterion, a MATLAB-controlled FEM batch analysis on COMSOL has been carried out to optimize pMUT’s multi-parameter geometry.

Study of Acoustic Matching Layers: Different from bulk piezoelectric transducers, MUT vibrates on the bending resonance mode of the thin diaphragm. Its process of transmitting and receiving ultrasound is more complicated, and cannot be applied with the traditional acoustic impedance and quarter wavelength matching theory [19,20]. Although, for pMUT transducers, there is necessity to have a matching layer between the vibration diaphragm and its transmission medium, and the acoustical matching problem has not been systematically studied in literature. Based on the round-trip models, we studied the influence of the acoustic matching layers of different density and wave velocity, especially layer thickness on pMUTs. The insertion of an acoustic matching layer can, in fact, be considered itself as a resonance layer modifying the diaphragm resonance frequency and bandwidth, and can be applied for all thin-film-based MUTs.

This work aims to present a new modeling method focusing on round-trip performance analysis of pMUT, introducing a more robust figure of merit—sensitivity fractional bandwidth product for transducers which is more suitable for structural optimization. The work implements a MATLAB—COMSOL joint simulation so as to calibrate the frequency shifting during a geometric parameter sweep, and gives guides on radius and thickness designation that traditional schemes have not clearly covered. Last but not least, this work proposes a bandwidth shaping technique using acoustic matching layers in pMUT for the first time, which could significantly increase the bandwidth.

## 2. Modeling and Analyzing Methods

The schematic diagram of the 10-MHz-centered pMUT element is shown in Figure 2, along with Table 1 showing initial geometry parameters. The pMUT element includes a cavity, above which is a supported compound plate which can resonate in its fundamental bending mode when actuated by applying an electric voltage at the upper deposited Aluminum Nitride (AlN) thin film (with Molybdenum electrodes at both sides). The realization of a pMUT array on substrate with cavities rather than deep through holes is for the reason that high frequency pMUT requires very small and precise radius to work at a desired resonant frequency, and etching shallow cavities (1–10 
μ
m in depth) is an easier approach [8,21] than etching deep through holes (over 200 
μ
m in depth) [22,23]. The above design structure of pMUT for modeling is referred to as cavity-pMUT.

### 2.1. Set-Up of Round-Trip Models

In order to obtain the round-trip characteristic of the cavity-pMUT and by taking into account the circular cavity element geometry, 2-D axisymmetric FEM models are set-up in frequency domain based on COMSOL Multiphysics 5.3a (COMSOL Inc., Burlington, Ma)
${}^{®}$
5.3a. Some previous works have tried to analyze the round-trip performance in time domain [24], and use Fast Fourier Transform (FFT) to simulate frequency response. However, the method is based on a model in which the excitation signal is a specific voltage pulse having limited bandwidth, and it is hard to get full band analysis or simulate a pulse-echo response under different waveform excitation. Here we establish frequency domain transmitting and receiving models separately, and use inverse Fast Fourier Transform (IFFT) to simulate a time domain pulse-echo response. Compared with round-trip analysis in time domain, calculating split FEM models in frequency domain is more stable, efficient and reusable.

Figure 3 shows the configuration of 2-D axisymmetric FEM transmitting model. The geometries are not in proportion in order to show each domain clearly. The substrate material is silicon in <100> orientation [25] (when using 2D axisymmetric model, the material will lose part of its anisotropic property in the rotation angular direction), the loading medium (fluid) is sunflower oil [26], Appendix A gives all the material properties used in the cavity-pMUT. The peripheral area of the model is an artificial domain, i.e., a perfectly matched layer (PML), where wave propagation is completely damped without reflection [27]. The “Typical Wavelength” parameter for PMLs in COMSOL, i.e., maximum stretched length, is set for the longest wavelength in both the loading medium and solid region—9000 [m/s]/Frequency. The outermost boundary (exterior edge of the PMLs) is “Low-Reflecting Boundary” for the solid region [28] and “Plane Wave Radiation” for the fluid region [29], respectively, to further enhance the wave absorption. At this same boundary, the electrical condition is the default “Zero Charge” one.

The receiving model possesses quite similar configurations with the former, but some modifications are obligatory. In the transmitting model, the depth of the load medium is 2 mm, the upper electrode is excited with a harmonic wave of unity amplitude in volt and zero phase. In the receiving model, the load medium has a depth of only 50 
μ
m, where the “Background Pressure Field” of unity amplitude in (Pascal) and zero phase is set, corresponding to a plane wave incidence towards the reception. The upper electrode is set to high impedance terminal (1 M
Ω
 load). The round-trip sensibility is then defined as the output voltage obtained at the upper electrode of the receiving model for a plane incident wave with its amplitude multiplied by the center far-field pressure (2 mm) of the transmitting model, as if the transmitting wave is reflected at its far-field, returned without loss and reconverted into voltage at the reception.

While studying the effect of acoustic matching layers, an additional solid material region is inserted between the load medium and the top electrode for both models, assigned with varying thickness, density, pressure-wave speed, shear-wave speed, and relative permittivity.

Figure 4 shows the general scheme of the meshing for the two FEM models. Firstly, all domains (not including PMLs) are meshed using the “Free Triangular” feature, constrained by maximum element size for each field: 
$c_{p}/\max\left(\mathrm{Frequency}\right)/4$
, in other words, 
min(λ)/4
, where 
$c_{p}$
 is the pressure-wave speed, and 
λ
 is the corresponding wavelength. The remaining domains, i.e., PML regions, are meshed using the “Mapping” feature that stretches the inner boundary mesh to the outer side of 30 elements in distribution.

### 2.2. Analysis of Round-Trip Performance

After obtaining both the transmitting and the receiving model results, the pressure at the far end of the load medium (2 mm), denoted as 
$S_{t}\left(\omega \right)$
, is the transmitting pressure sensitivity of the pMUT, and the voltage response at the receiving electrode, denoted as 
$S_{r}\left(\omega \right)$
, is the receiving voltage sensitivity. The round-trip sensitivity is then determined by multiplying 
$S_{t}\left(\omega \right)$
 and 
$S_{r}\left(\omega \right)$
, seen in Equation (Equation 1):
(1)
$$F_{pmut}\left(\omega \right)=S_{t}\left(\omega \right)\cdot S_{r}\left(\omega \right)$$


To simulate the time domain response, we first calculate the frequency spectrum of an arbitrary excitation waveform (in time domain), and multiply it with calculated round-trip sensitivity 
$F_{pmut}\left(\omega \right)$
. The resulting frequency spectrum is then inversely transformed into time domain using inverse Fast Fourier Transform (IFFT), seen in Equation (Equation 2).

(2)
$$f_{response}\left(t\right)=F^{-1}\left\{F\left[f_{pulse}\left(t\right)\right]\cdot F_{pmut}\left(\omega \right)\right\}$$


The product of round-trip sensitivity and relative bandwidth (SBW), is calculated by Equation (Equation 3), where 
$BandWidth_{-6dB}$
 is the −6 dB bandwidth of the round-trip sensitivity 
$F_{pmut}\left(\omega \right)$
, 
$f_{c}$
 is average value of the upper and lower bounds of −6 dB bandwidth.

(3)
$$\mathrm{SBW}=max\left(abs\left(F_{pmut}\left(\omega \right)\right)\right)\cdot BandWidth_{-6\;\mathrm{dB}}/f_{c}$$


SBW is used as the criterion for optimization in this work, since it represents the global transmitting-receiving sensitivity and bandwidth in a loaded condition rather than the device resonance in a vacuum (as electromechanical coefficients assumes [13,14]), and it also considers the whole process from excitation to reception. Besides, the sensitivity and bandwidth are mutually compromising factors, thus making SBW a relatively stabilized coefficient. SBW is widely applicable and comparable, since both bulk piezoelectric transducers and MUTs use pulse-echo experiments as an important means of characterization [30,31].

### 2.3. MATLAB-Controlled Parameter Sweeping

Based on the round-trip models and analyzing methods, we can sweep parameters to optimize the geometries, with a target center frequency 
$f_{t}$
 = 10 MHz, for example. In this part we use LiveLink
${}^{\mathrm{TM}}$
 for MATLAB
${}^{®}$
 that integrates COMSOL Multiphysics
${}^{®}$
 with MATLAB
${}^{®}$
 scripting to accelerate the computation and process data in batches. Figure 5 shows flow chart of the MATLAB program.

In the following, four pMUT geometrical parameters (represented in vector 
X
) are first optimized: the top electrode radius 
$r_{t}$
, the piezoelectric thin film radius 
$r_{p}$
 and thickness 
$t_{p}$
, and the vibration diaphragm thickness 
$t_{s}i$
. The optimization of radius for both top electrode and piezoelectric layer has yet not been systematically studied, since the stiffness of the vibration diaphragm changes a lot when the piezoelectric layer radius varies, resulting in uncontrollable center frequency.

In general, the program opens the transmitting model and receiving model in sequence, edits the models’ parameters and executes the calculation in COMSOL’s kernel, then extracts the required data to calculate 
$F_{pmut}\left(\omega ,X\right)$
 and SBW. As 
$f_{c}$
 is the center frequency of −6 dB bandwidth of 
$F_{pmut}$
, it can be changed when the geometrical parameters are swept in the optimization process. So in the program, a switch loop is used to adjust the cavity radius 
$r_{c}$
 in a way that 
$f_{c}$
 always meets the target frequency 
$f_{t}$
. According to clamped plate’s vibrating theory [32], the pMUT resonance frequency 
$f_{r}$
 can be estimated by the following relation:
(4)
$$f_{r}\propto t/r^{2}$$

where *t* is the plate’s effective thickness, 
$r_{c}$
 is the plate’s effective radius.

The program adjusts the current cavity radius 
$r_{c}$
 by multiplying it with a factor 
$\sqrt{f_{c}/f_{t}}$
 while keeping *t* constant, it continues to calculate until the 
$f_{c}$
 reaches 
$f_{t}$
 within an error of 1%. Such a frequency correction by 
$r_{c}$
 compensation is considered as a frequency normalization or the frequency calibration as we call it in the following.

## 3. Results

### 3.1. Analysis of Round-Trip Models

The transmitting model and receiving model have been solved separately. Figure 6a shows a FEM calculation result obtained with parameters: 19 
μ
m top electrode radius, flat piezoelectric layer (100 
μ
m radius) and 30.9 
μ
m cavity radius. The transmitting sensitivity has a peak value of 75.55 Pa/V at 10.6 MHz, and the receiving sensitivity has a peak value of 4.19 
μ
V/Pa at 9.2 MHz. By multiplying two sensitivity functions together results the round-trip sensitivity, seen Figure 6b. The absolute round-trip response of the pMUT shows a 40% relative bandwidth and a maximum sensitivity of 
$2.83\times 10^{-4}$
, i.e., −71 dB with 
$f_{c}$
 occurring at 9.94 MHz.

It should be noted that the peak of the far field pressure sensitivity (left axis in Figure 6a) is different from the receiving sensitivity. This is normal due to their intrinsic relationship. For instance, the far field pressure response 
$p_{f}$
, as mentioned, results from the integral of surface acoustic pressure, and the acoustic pressure of each surface point follows boundary condition of acoustic-structure interaction:
(5)
$$-n\cdot \left(-\frac{1}{\rho_{c}}\left(▽p_{t}-q_{d}\right)\right)=-n\cdot u_{\mathrm{tt}}$$


(6)
$$F_{A}=p_{t}\cdot n$$

where 
$u_{\mathrm{tt}}$
 is the structural acceleration, 
n
 is the surface normal, 
$p_{t}$
 is the total acoustic pressure, 
$q_{d}$
 is the dipole domain source (if applicable) and 
$F_{A}$
 is the load (force per unit area) experienced by the structure. Here, the structural acceleration 
$u_{\mathrm{tt}}=\omega^{2}u$
 contains the quadratic term of the frequency. The peak frequency of far field pressure response 
$p_{f}$
 then shifts due to dispersion effect. Additionally, the higher-order mode is more evident in the far field pressure sensitivity response, as seen in Figure 6a.

Next, a square-wave pulse of 16.67 ns duration and 1V amplitude is used to simulate the pulse-echo response, its frequency spectrum has a first zero point at 60 MHz (Figure 7a), multiplying the spectrum of the pulse by the round-trip sensitivity results in the pulse-echo spectrum (shown in black dotted line). Figure 7b shows the simulation result in the time domain. The pMUT element’s response of the square-wave pulse presents an enveloped sine wave pattern lasting for about five cycles, which indicates that the bandwidth of the pMUT is not large enough in this situation.

### 3.2. Optimization Study of Radius

The optimization starts with the validation of traditional pMUT design [13], in which the top electrode radius is first optimized. The program sweeps the top electrode radius from 12 
μ
m to 36 
μ
m, and the initial cavity radius is 30 
μ
m. By adjusting the cavity radius, the program holds 
$f_{c}$
 at the target center frequency of 10 MHz, seen Figure 8.

It should be noted that, in a previous study [13], the effect of varying top electrode radius on center frequency is neglected; when the top electrode radius reaches the cavity radius, the electromechanical coefficient turns to be zero, with such result concluded from the analytical theory of the equivalent plate. However, the FEM modeling considers the real boundary condition, in which pMUT can still work when the radius exceeds, with working frequency shifted for over 10%. The calibrated result shows that SBW reaches a maximum of −79 dB when the top electrode radius is 61.60% of the cavity radius.

To optimize the radius of both top electrode and piezoelectric layer, the calibration process is mandatory since the center frequency shifts more. In this work, the programmed calibration process accelerates the optimization, reduces computational complexities, and solves the problem of non-converging frequency.

The results are plotted as pseudo-color images in Figure 9a, where each pixel point represents a swept 
$r_{p}$
 and 
$r_{t}$
 radii with the working frequency 
$f_{c}$
 convergent at 10-MHz-target-frequency by adjusting the cavity radius 
$r_{c}$
. The corresponding 
$r_{c}$
 data is shown in Figure 9b. The maximum SBW is very close to the hypotenuse of the triangular figure, where 
$r_{t}$
 = 21.0 
μ
m, 
$r_{p}$
 = 23. 2 
μ
m with 
$r_{c}$
 = 30.9 
μ
m or 
$r_{t}/r_{c}=68\%$
, and 
$r_{p}/r_{c}=75\%$
. The 2 
μ
m difference between 
$r_{t}$
 and 
$r_{p}$
 is probable due to effect of the load medium’s dielectric field. In the actual fabrication process, this difference can be neglected, since etching the piezoelectric layer by self-alignment (where 
$r_{t}=r_{p}$
) can save an additional group of lithography processes.

Figure 10 compares the performance of the traditional design of 10 MHz pMUT and radii-optimized pMUT. Though the bandwidth is lowered by only 6%, the maximum round-trip sensitivity has been improved by 61%, resulting 52% improvement in total SBW. The same trend can also be drawn from the time domain response.

Another performance indicator frequently used for ultrasonic transducers is—the electromechanical coupling factor 
$k_{eff}^{2}$
. To compare it with the SBW optimized pMUT, we have calculated the pMUT transmitting models in an air load medium, and extracted the admittance frequency response as shown in Figure 11. Since the load medium is air, the vibration of the pMUT is nearly lossless, and the maximum admittance frequency 
$f_{m}$
 and the minimum admittance frequency 
$f_{n}$
 are both increased to about 16 MHz.

The electromechanical coupling factor 
$k_{eff}^{2}$
 is usually determined by [33],

(7)
$$f_{eff}^{2}=\frac{f_{p}^{2}-f_{s}^{2}}{f_{p}^{2}}$$

where 
$f_{p}$
 is parallel resonance frequency and 
$f_{s}$
 is motional (series) resonance frequency. These two characteristic frequencies can be approximated to 
$f_{n}$
 and 
$f_{m}$
, respectively, when losses are small. The 
$k_{eff}^{2}$
 of the traditional design pMUT and radii-optimized pMUT are 3.08% and 2.50%, respectively. However, the SBW performance has actually been improved for over 50% in the latter case as Figure 10 indicates. The 
$k_{eff}^{2}$
 is a good performance evaluation parameter for resonators, but is not convenient for heavily loaded ultrasonic transducers, or in pMUT design and optimization.

### 3.3. Optimization Study of Thickness

When the thickness changes, the radius ratio of the designed pMUT is retained in this work since the coupling between the radius and the thickness is very limited. While looking into conventional analytical models, the process of analyzing the resonant frequencies or the electromechanical coupling coefficient usually dissociates the radius and thickness for different parts of the equations [9,13], where the thickness parameter is integrated into the diaphragm’s bending stiffness, and the relative radius of the top electrode to the cavity radius reflects how the electric field is distributed along the radius.

The essence of the classical optimization scheme is to apply the strain in a way that excites the resonance vibration with maximum efficiency, where the top electrode radius is being tuned. In this work, the optimization of the piezoelectric layer radius aims to do the same, which is to further concentrate the stress response. On the basis of this, and considering the huge computation resources required for full-parameter sweep, the optimizations are carried out separately.

Thickness optimizations are carried out with the radius ratio of the electrode and piezoelectric layer to the cavity kept at the optimum: 
$r_{t}/r_{c}=68\%$
 and 
$r_{p}/r_{c}=75\%$
. Considering the available fabrication process, the thickness of AlN—
$t_{p}$
 (fabricated by magnetic sputtering) is swept from of 0.2 
μ
m to 2 
μ
m, the thickness of passive diaphragm in Si—
$t_{si}$
 (fabricated by transfer-release process of SOI’s device layer) is swept from of 0.5 
μ
m to 5 
μ
m.

As has been stated in Equation (Equation 4), the change in total thickness 
$t_{total}=t_{p}+t_{si}$
 will significantly change the resonant frequency of pMUT, so that the cavity radius calibration is applied as well. Figure 12 shows calculated SBW by varying 
$t_{p}$
 and 
$t_{si}$
. The maximum SBW is located at 
$t_{p}=2$

μ
m and 
$t_{si}=5$

μ
m which are the maximum values both for 
$t_{p}$
 and 
$t_{si}$
 in their optimization sweep. This means that SBW can still be increased if we continue to increase 
$t_{p}$
 and 
$t_{si}$
.

However, in such circumstances, the cavity radius 
$r_{c}$
 is also changed to 35 
μ
m, while its initial value is 30 
μ
m, as seen in Figure 12b. Another question arises here, that 
$r_{c}$
 cannot be changed too much in a pMUT array since it is limited by the array pitch. Moreover, for a fixed pitch size, the modification in 
$r_{c}$
 will change the active vibration area, hence having a different transmission energy ratio and effective sensibility. For this reason, we make a correction in the SBW definition by normalizing it as SBW by unity active surface area, which is SBW/
$S_{a}$
, with 
$S_{a}=\pi r_{c}^{2}/\pi r_{c0}^{2}$
 and 
$r_{c0}$
 the initial cavity radius of 30 
μ
m. For most ultrasonic applications, the unit area sensitivity (Normalized SBW) is the most valuable criterion, since the performance of the entire transducer can be improved by connecting multiple elements of high area efficiency.

With such modification, the maximum SBW is at 
$t_{si}$
 = 0.5 
μ
m and 
$t_{p}$
 = 0.4 
μ
m as shown in Figure 13. The trend has been inverted such that the reduction of diaphragm thickness brings increased SBW by unity active area. Normally for a SOI wafer with a 4 
μ
m thickness device layer, the optimal AlN thickness should be 1.4 
μ
m to have a maximum SBW of −74 dB.

The optimal thickness ratio of 
$t_{p}/t_{si}$
 can be derived from the figure above, which is 35∼40%, as shown in Figure 14. However, it should be noted that the step sizes for thickness optimization are divided equally during the sweep, which leads to difficulties in retrieving the optimal ratio accurately for pMUT of very thin passive diaphragm.

The problem emerges when selecting the optimal parameters for the pMUT, e.g., a 0.5 
μ
m thickness passive diaphragm and a 0.2 
μ
m thickness piezoelectric layer. To fabricate Cavity-SOI with only a 0.5 
μ
m device layer, serious stress problems will be encountered during the bond-release process, and lead to severe collapse and decreased yields. Besides, for pMUT deposited with a 0.2 
μ
m thickness piezoelectric layer, the operating voltage range will drop significantly to tens of volts, due to the limited breakdown strength of the piezoelectric film, which will limit its application areas, especially for medical imaging and non-destructive testing.

For pMUTs of the same operating frequency, less total thickness can reduce the bending stiffness, and makes the diaphragm more prone to bending. Therefore, designers need to make a compromise between performance, process yield and operating voltage range required for the target application, and then choose the thinnest available device layer thickness to achieve maximum performance.

### 3.4. Study of Acoustic Matching Layers on pMUTs

To study the effect of matching layers on the pMUT, we calculated the round-trip FEM models in a frequency dimension of 0.2∼40 MHz and a matching layer thickness of 2∼100 
μ
m, with different materials including polydimethylsiloxane (PDMS), polyurethane, and high-density polyethylene (HDPE). The properties of the materials are listed in Table A5. The materials that are mostly used in bulk piezoelectric transducers, including AAO–epoxy, Epotek 301, aluminium, metal oxide, etc. [34], are not options for MUTs since their acoustic impedance is too high to allow thin films to vibrate.

The acoustic impedance 
$Z_{M}$
 of PDMS, polyurethane and HDPE are 1.05, 1.42, and 2.22 MRayl, respectively, based on Equation (Equation 8), where 
$\rho_{M}$
 represents the density and 
$c_{p}$
 the compression wave speed. Besides, PDMS and polyurethane are nearly incompressible (Poisson’s ratio 
σ≈0.5
), so their shear-wave speed are accounted as zero in the models.

(8)
$$Z_{M}=\rho_{M}\times c_{p}$$


The polyurethane has nearly no effect on the sensitivity and bandwidth, since the acoustic impedance is very close to the load medium (1.6 MRayl). It acts like a sound transmission layer and the calculation result is omitted. The study results of PDMS and HDPE are shown in Figure 15 where the pseudo-color level represents the round-trip sensibility but not the SBW to well analyze the bandwidth property.

There are two interesting ranges of thickness with PDMS (Figure 15a), which help expanding the bandwidth, including 25∼30 
μ
m and 67∼72 
μ
m. At the thickness of 25 
μ
m the pMUT has the best relative bandwidth of 104% (
$f_{c}=14.92$
 MHz) but the maximum round-trip sensitivity of −74 dB is found at PDMS thickness of 67 
μ
m with a relative bandwidth of 69% compared with that without matching layer (
rBW=38%
). For pMUTs with HDPE matching layers (Figure 15b) there are too many loss valleys in the frequency response due to shear-wave resonances at its odd multiples of 1/4 wavelength, making devices of very low bandwidth. Thus HDPE might not be a good choice for the acoustic matching of the pMUT.

From the PDMS pseudo-color map, it can be found that the compression-wave related resonance peaks and valleys appear periodically by 1/4 wavelength. The large bandwidth does not occur at the thickness where the compression-wave fundamental resonance happens, i.e., 10 MHz, but at the transient thickness region where the fundamental resonance and the second order resonance have nearly the same peak level, i.e., the 25 
μ
m thickness and 67 
μ
m thickness in Figure 15a.

Finally, the sensitivity performance and pulse-echo simulation are calculated and shown in Figure 16. The pMUTs with matching layers both show compromised sensitivity in Figure 16a, however they all have expanded bandwidth, of 69% and 104% for the 67 
μ
m thickness and 25 
μ
m thickness PDMS matching, respectively. Figure 16b gives a time-domain evaluation based on the excitation of a 16.67 ns width square-wave pulse. The pMUT with 67 
μ
m PDMS shows a shorter trailing wave with slightly lowered amplitude, where the pMUT with 25 
μ
m PDMS shows a surprisingly short pulse length, that the pulse damps quickly after the first cycle due to higher bandwidth and center frequency (which is 8.22 MHz *BW* with 11.97 MHz 
$f_{c}$
, and 15.51 MHz *BW* with 14.92 MHz 
$f_{c}$
, respectively).

### 3.5. Fitting and Re-Optimization

In order to demonstrate the validity and universality of the main optimization scheme, the previous pMUT works of classical structures are reviewed and further optimized in the dimensions of top electrode radius and piezoelectric radius. Figure 17 shows the fitting and optimizing procedure based on the design and experiment result of the prior works. Since the center frequency of the device is more sensitive to the variation in the cavity radius 
$r_{c}$
 (from lithography deviation and imperfect etching process), we treat 
$r_{c}$
 as a main factor that shifts the frequency, and fit it based on electron microscopy of cross sections (if available) or original design parameters within an appropriate deviation range.

Figure 18 shows the geometry schematic for each structure of pMUT. Corresponding FEM models have been created based on the characterized/designed geometries in those works, with similar settings of physical field, boundary conditions and mesh divisions as shown in Figure 3 and Figure 4. Taking into account the process deviation and the material damping, the cavity radius 
$r_{c}$
, 
$e_{31,f}$
 coefficient, and loss factor 
$\eta_{p}$
 of the piezoelectric layer are adjusted so that the sensitivity-bandwidth product gets close to the measurement result. By fitting the experiment data, we set up a group of basic models as a starting point for optimization.

Next, the top electrodes and piezoelectric layers in the above models are applied with the proposed optimization scheme, where 
$r_{t}=r_{p}=75\%r_{c}$
, and 
$r_{c}$
 is tuned to the same operating frequency. Table 2 compares the experimental, fitted and the optimized result, including center frequency, sensitivity, relative bandwidth and sensitivity—relative bandwidth product of referred pMUTs for 25 MHz AlN-pMUT (with and without sealing material Parylene-C), 20 MHz AlN-pMUT and 11 MHz PZT-pMUT, respectively.

The SBW of all four fitted pMUT models shows good coherence to the LDV result. Suppose that the actual thickness of each layer can be extracted from scanning electron microscope (SEM), the *e
${}_{31,f}$
* can be fitted more accurately and as a good reference for further pMUT design. Finally the last four columns of Table 2 present the calculation result of the optimized models. SBW merits of all devices are enhanced, such as from 7 pm/V to 70 pm/V for the first one, meaning that the top electrode design is much disadvantaged. The SBW merit of the last two devices shows 50% enhancement, since the radii of top electrodes have already been optimized based on analytical methods.

## 4. Summary and Outlook

In this work, we first established a FEM model that can evaluate the round-trip performance of cavity structure pMUT in both frequency and time domain. The frequency spectrum of round-trip sensitivity and simulated pulse-echo waveform under wide-band excitation give the most quantitative and intuitive feedback for pMUT design. Based on the calculated round-trip response and bandwidth, their product is defined as SBW which is used later as a performance criterion in the structure and parameter optimization of the MUTs.

Secondly, the geometries of pMUTs, including the radius of the top electrode and piezoelectric layer, and thickness of piezoelectric film and vibration diaphragm, have been optimized. We used the SBW optimized pMUT parameters to estimate the traditional performance indicator—electro-mechanical coefficient 
$k_{eff}^{2}$
, showing that 
$k_{eff}^{2}$
 is not sensitive to the pMUT geometrical change, and cannot give good indication of transmission and reception sensitivity at the same time, especially in the case when the pMUT is loaded with a transmission medium.

Since the center frequency of the pMUTs can shift due to the geometry optimization, the pMUTs cavity radius is adjusted to compensate the frequency shift, keeping the working frequency at the design target one. We have set-up an efficient program based on LiveLink
${}^{\mathrm{TM}}$
 for MATLAB
${}^{®}$
 to accelerate the modeling process during such optimization.

The result of geometry optimization shows a 52% enhancement of SBW when the top electrode and piezoelectric layer are both etched to 75% radius of the cavity beneath. It also reminds designers to make a compromise between performance, process yield and operating voltage range required for the target application, and then choose the thinnest available device layer thickness to achieve maximum performance.

In the last part, we studied the effect of the acoustic matching layer on the pMUT performance with several possible matching materials. As the traditional matching criterion for bulk piezoelectric transducers is not applicable for flexural vibration pMUTs, the pMUT resonance behavior is determined by calculating the round-trip response in a broad frequency and thickness range of the matching layer. Using PDMS as the matching material, or another material with a similar acoustic impedance, larger bandwidth can be obtained for a layer thickness near its odd multiple of quarter wavelength, but the maximum round-trip sensitivity occurs at its multiple of half wave-length. The rubbers and some incompressible polymers of low acoustic impedance can be the potential candidates of acoustic matching layers for pMUTs.

## Figures and Tables

**Figure 1 sensors-22-02307-f001:**
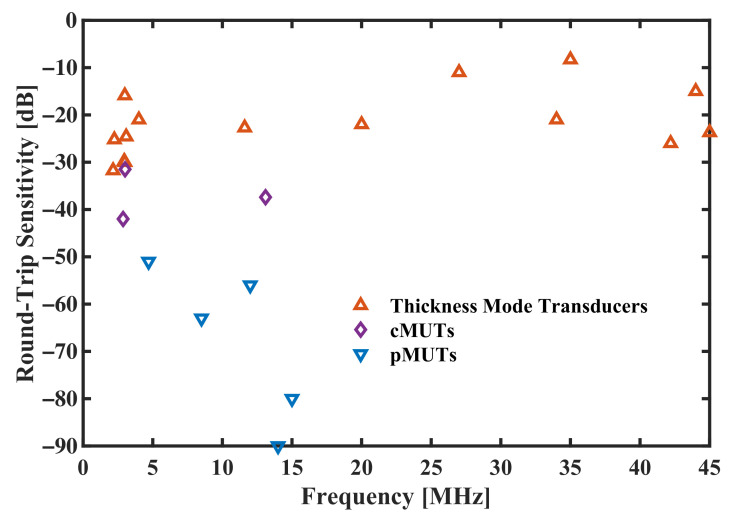
Round-trip sensitivity comparison between pMUTs, cMUTs and thickness mode transducers.

**Figure 2 sensors-22-02307-f002:**
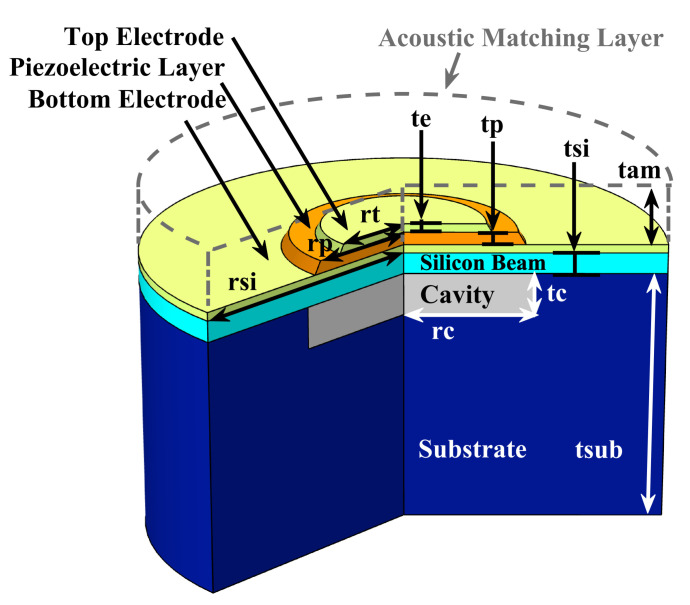
3-D schematic diagram of the designed cavity-pMUT.

**Figure 3 sensors-22-02307-f003:**
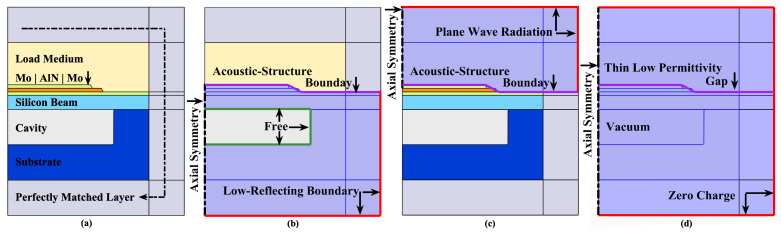
2-D axisymmetric configurations, showing (**a**) Materials and PMLs. (**b**) Domain and boundary conditions of solid mechanics. (**c**) Domain and boundary conditions of pressure acoustics. (**d**) Domain and boundary conditions of electrostatics.

**Figure 4 sensors-22-02307-f004:**
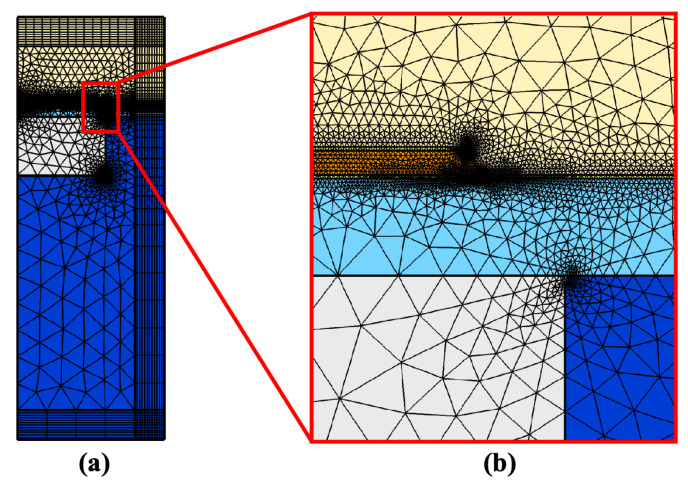
Meshing scheme diagram of the 2-D axisymmetric models. (**a**) Overall meshing situation (**b**) Meshing situation around the diaphragm.

**Figure 5 sensors-22-02307-f005:**
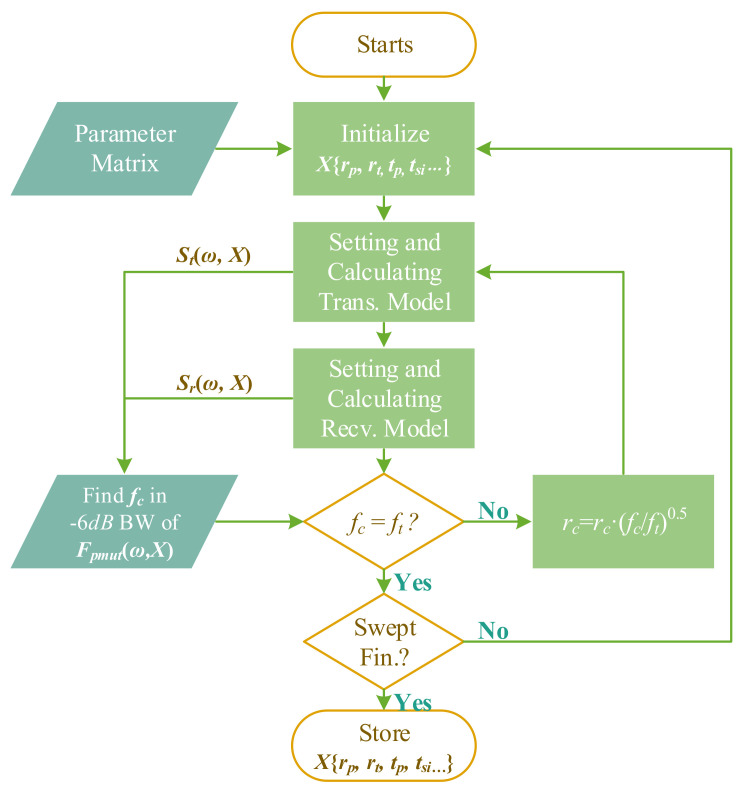
Program flow of LiveLink
${}^{\mathrm{TM}}$
*for* MATLAB
${}^{®}$
 for optimizing pMUTs’ geometries.

**Figure 6 sensors-22-02307-f006:**
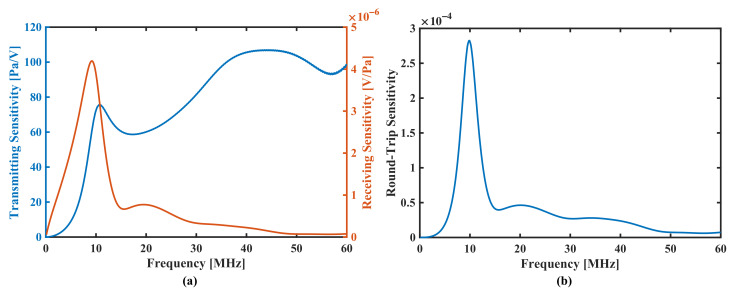
Frequency response of sensitivities. (**a**) the blue line shows an absolute acoustic pressure at 2 mm center by unit voltage excitation; the orange shows absolute receiving voltage by unit backing pressure. (**b**) shows the resulting round-trip sensitivity.

**Figure 7 sensors-22-02307-f007:**
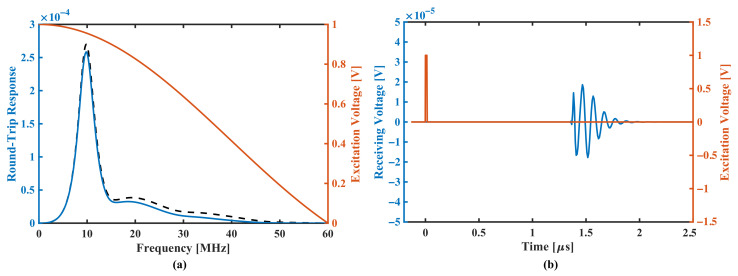
Pulse-echo simulation results, where blue line denotes the pMUT element and orange line denotes the square-wave pulse. (**a**) shows absolute frequency spectrum of the pulse, the round-trip response and the reception spectrum (dotted line). (**b**) shows time-domain waveforms, respectively.

**Figure 8 sensors-22-02307-f008:**
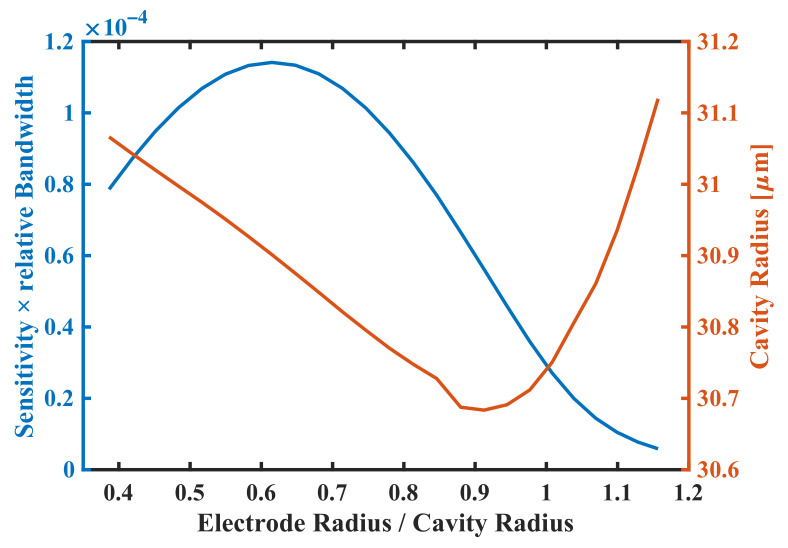
Blue line shows SBW performance of traditional pMUT design of varying top electrode radius, and orange line shows cavity radius calibrated for pMUT to working at 
$f_{c}$
 of 10 MHz.

**Figure 9 sensors-22-02307-f009:**
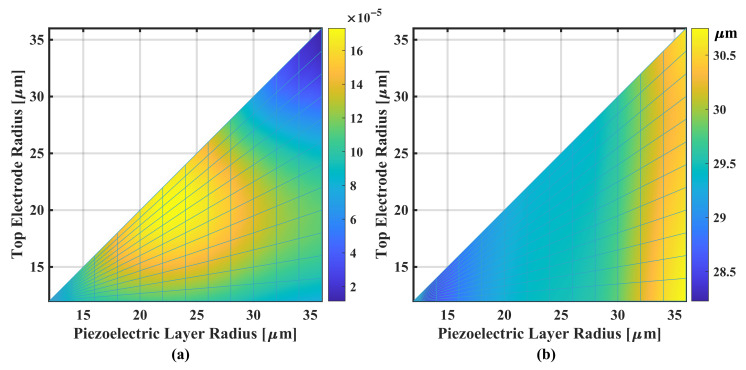
Pseudo-color plot of (**a**) the SBW of varying 
$r_{p}$
 and 
$r_{t}$
 and (**b**) the corresponding 
$r_{c}$
 used that fixes 
$f_{c}$
 at 10 MHz.

**Figure 10 sensors-22-02307-f010:**
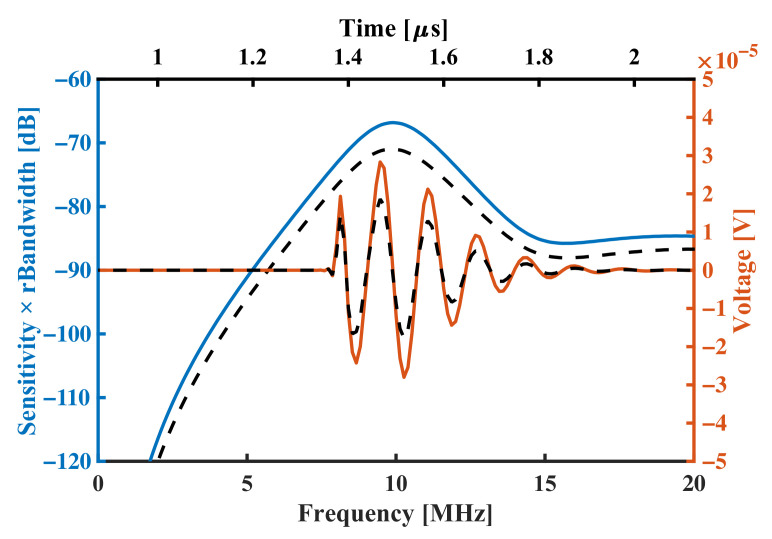
Frequencyspectrums (**left axis**) and pulse-echo waveforms (**right axis**) comparing radii-optimized pMUT (**solid lines**) with traditional pMUT design (**dashed lines**).

**Figure 11 sensors-22-02307-f011:**
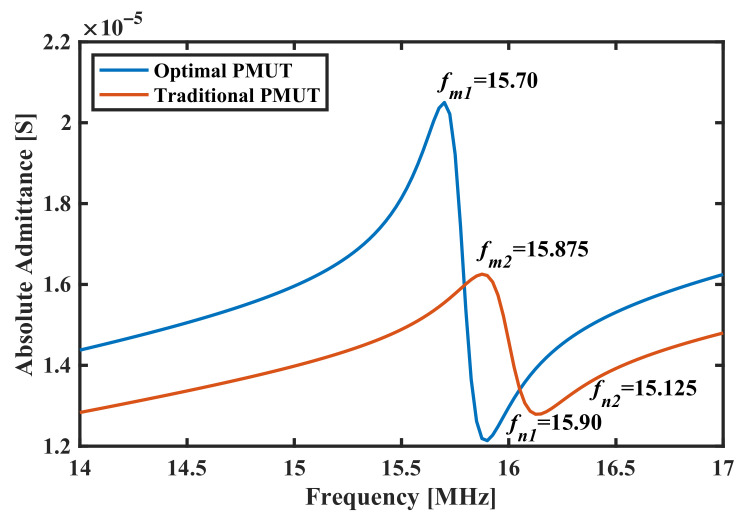
Frequency responses of admittance comparing radii-optimized pMUT (**blue line**) with traditional pMUT design (**orange line**).

**Figure 12 sensors-22-02307-f012:**
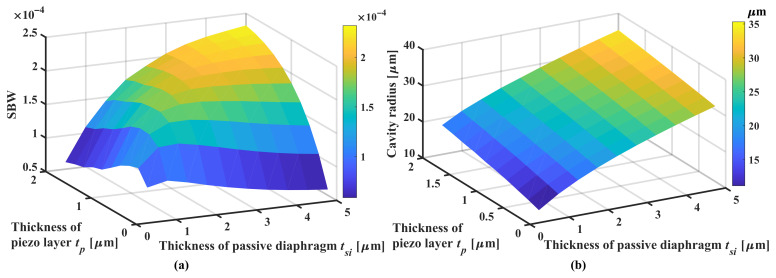
(**a**) SBW performance of the pMUT with varying 
$t_{p}$
 and 
$t_{si}$
. (**b**) Cavity radius used for each thickness combination to converge the center frequency.

**Figure 13 sensors-22-02307-f013:**
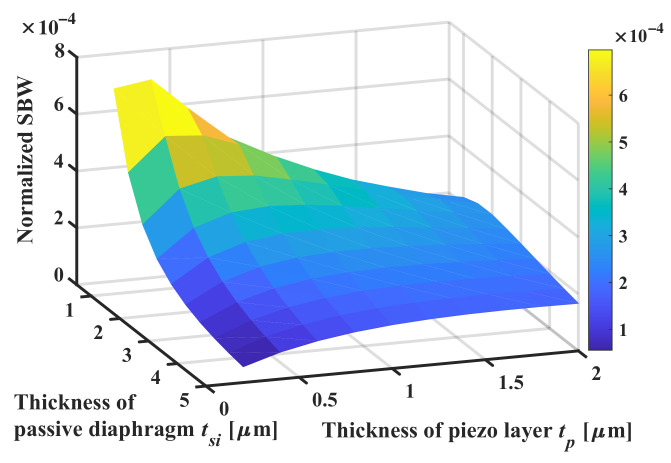
SBW normalized by active surface area of different 
$t_{p}$
 and 
$t_{si}$
.

**Figure 14 sensors-22-02307-f014:**
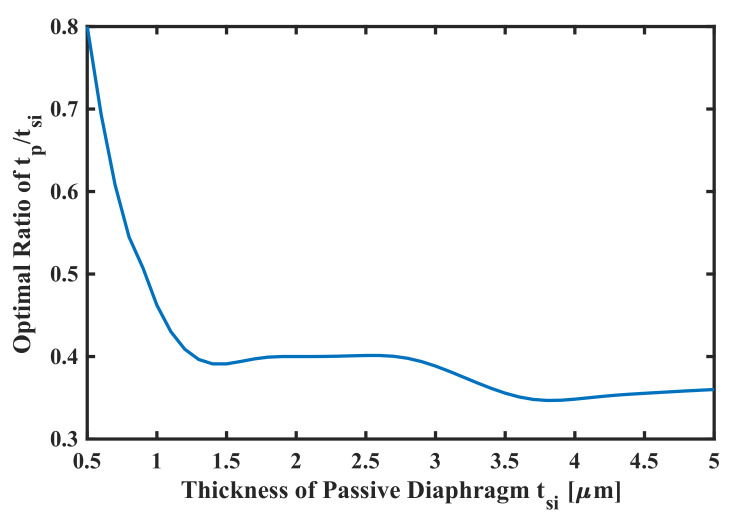
Optimal ratio of 
$t_{p}/t_{si}$
 by sweeping thickness of the passive diaphragm.

**Figure 15 sensors-22-02307-f015:**
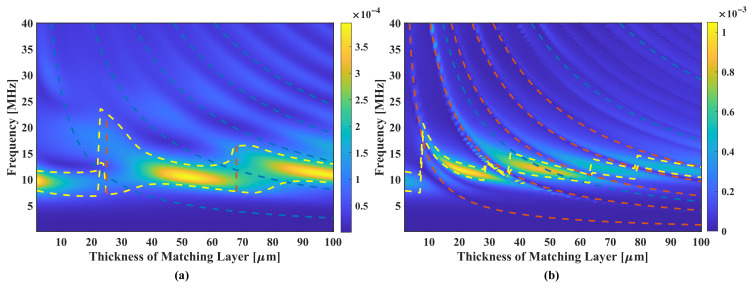
Round-trip sensitivity of models with matching layers of varying thickness. (**a**) shows models with PDMS layers; (**b**) shows models with HDPE layers. In the pseudo-color plot, the colormap represents the amplitude of sensitivity; the blue (compression-wave related) and orange (shear-wave related) dashed lines represent thickness-frequency relationship with a specific wavelength ratio; the yellow dashed lines represent upper and lower bounds of −6 dB bandwidth.

**Figure 16 sensors-22-02307-f016:**
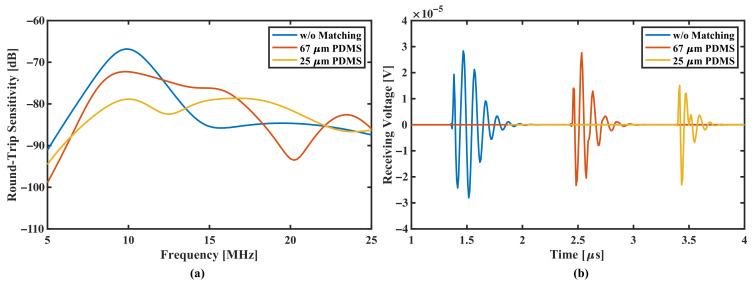
(**a**) Frequency spectrum of round-trip sensitivity and (**b**) time domain pulse-echo response of a square-wave pulse of 16.67 ns width, comparing radii-optimized pMUTs without matching layer, with 67 
μ
m thickness PDMS layer and with 25 
μ
m thickness PDMS layer.

**Figure 17 sensors-22-02307-f017:**
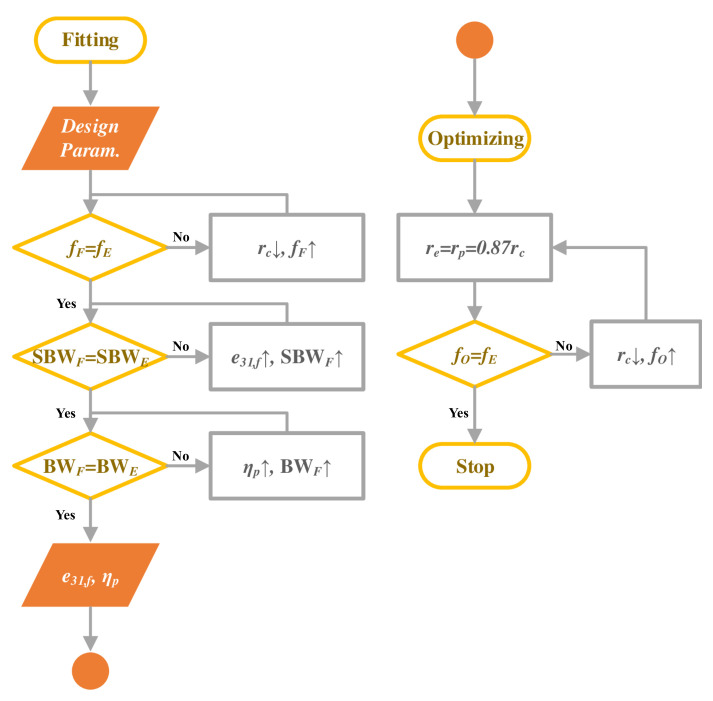
Flowchart of fitting and optimizing process. The subscript ’E’ stands for ’Experiment’, ’F’ stands for ’Fitting’, and ’O’ stands for ’Optimizing’.

**Figure 18 sensors-22-02307-f018:**
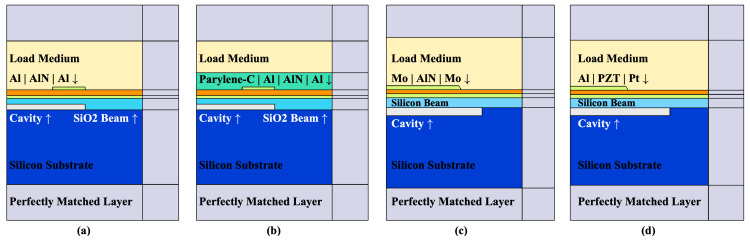
Cross-sectional 2-D axisymmetric diagram of the models of the prior works. (**a**) AlN-pMUT working at 25 MHz realized by releasing sacrificial layers [35]; (**b**) AlN-pMUT working at 25 MHz realized by releasing sacrificial layers, with 3 
μ
m Parylene-C [35]; (**c**) AlN-pMUT working at 20 MHz realized by using Cavity-SOI wafers [36]; (**d**) PZT-pMUT working at 11 MHz realized by using Cavity-SOI wafers [8].

**Table 1 sensors-22-02307-t001:** Geometry parameters used for designed pMUT element.

Structure	Material	Thickness ( μ m)	Radius ( μ m)
Top Electrode	Molybdenum	$t_{e}$ 0.15	$r_{t}$ 20
Piezoelectric Layer	Aluminum Nitride	$t_{p}$ 0.80	$r_{p}$ 22
Bottom Electrode	Molybdenum	$t_{e}$ 0.15	$r_{si}$ 100
Passive Diaphragm	Silicon	$t_{si}$ 4.00	$r_{c}$ 29.35
Cavity	Vacuum	$t_{c}$ 5.00	$r_{c}$ 29.35
Substrate (half-pitch)	Silicon	$t_{sub}$ 100	$r_{si}$ 100

**Table 2 sensors-22-02307-t002:** Design parameters, fitting parameters and optimization results of the reviewed pMUTs.

	Data Type	$r_{c}$ ( μ m)	$e_{31,f}$ (C/m ${}^{2}$ )	$\eta_{p}$	Frequency (MHz)	Sensitivity (nm/V)	Relative Bandwidth	SBW (pm/V)
25 MHz AlN-pMUT air-loaded	Experiment Fitting Optimized	12.5 13.70 12.32	- 0.42 0.42	- 0.0023 0.0023	25 24.99 25.01	2.5 2.55 167.51	0.28% 0.27% 0.04%	7.02 6.99 70.32
23 MHz AlN-pMUT +Parylene-C	Experiment Fitting Optimized	12.5 13.01 12.15	- 0.42 0.42	- $\eta_{p}$ = 0.0023 $\eta_{pc}$ = 0.033	23.45 23.45 23.46	0.36 0.36 2.56	1.14% 1.14% 1.48%	4.09 4.04 37.76
20 MHz AlN-pMUT air-loaded	Experiment Fitting Optimized	25.0 23.90 22.30	- 1.47 1.47	- - -	20 19.97 20.02	10 9.61 20.26	1.00% 1.03% 0.74%	100 99.29 149.32
11 MHz PZT-pMUT air-loaded	Experiment Fitting Optimized	25.0 25.75 24.4	- 6.60 6.60	- 0.007 0.007	11.06 11.05 11.05	316 323.23 545.06	0.45% 0.44% 0.39	1427 1431 2108

$\eta_{pc}$
 used here is the loss factor for sealing material Parylene-C, during fitting of which the loss factor of piezoelectric material is fixed and inherited from the upper model without Parylene-C, where 
$\eta_{p}$
 = 0.0023.

## Data Availability

Data sharing not applicable.

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
