# Peer review of "Sensitivity—Bandwidth Optimization of PMUT with Acoustical Matching Using Finite Element Method"

_sensors, 2022, doi:10.3390/s22062307_

Round 1
Reviewer 1 Report
The manuscript presents a sensitivity-bandwidth optimization of pMUTs using FEM simulation. Overall, the work is good. There are a few suggestions that might help improve the paper:
- The work conducts the radius optimization and thickness optimization separately. In line 266-267, the authors claim the thickness optimization was carried out with optimized radius ratio. However, radius optimization and thickness optimization are coupled, especially when cavity radius is changed accordingly to maintain the frequency. Could the authors clarify and explain whether the thickness change will affect the radius ratio optimization?
- In Fig. 5a, the transmitting sensitivity doesn’t peak at the resonant frequency, but varies and increase with the frequency. Please give explanation for this result.
- 11 and Fig. 12 give different trends. Which one should the designer chase to optimize the thickness? The thickness optimization doesn’t give a clear conclusion or guidance to the design.
- Does the author consider the thickness ratio of the piezoelectric layer and passive diaphragm in the optimization? It will directly affect the neutral plane, electromechanical coupling and sensitivity.
- In line 348-350, “… should be as thin as possible…” . This is confusing, please explain and give evidence to support this conclusion.
Reviewer 2 Report
This work extensively presents some fundamental investigations of PMUT designs, however, most of the content is well known in the field. It can be improved by at least having comparisons with prior arts of PMUT research such as the extension of the reference papers. There are many nonrelevant papers as references such as COMSOL user guide, Timoshenko, or BVD model. The authors should take a similar FoM (sensitivity &bandwidth) to look into with the PMUT prior arts.
Reviewer 3 Report
Objective: A new model in finite element method (FEM) to study round-trip performance of piezoelectric micromachined ultrasonic transducers (pMUTs) is proposed.
Material and methods: The round-trip sensitivity of pMUT is defined as the product of the frequency response of transmitted far field pressure to source voltage excitation and that of reception output to return wave pressure.
A multi-parameter optimization for a cavity pMUT is performed based on the sensitivity-bandwidth product considering the Sensitivity and fractional BandWidth (SBW) performance indicator.
The radii of the electrode and the piezoelectric layer, the thicknesses of the piezoelectric layer and the vibration diaphragm are adjusted in a way to obtain the largest SBW.
Novelty: an acoustic matching method is proposed and applied to pMUTs for the first time.
Results: The optimization enhances the SBW by 52% when the top electrode and piezoelectric layer are both etched to 75% radius of the cavity beneath.
Conclusion: the introduction of acoustic matching layer shows
significant bandwidth expansion in both transmitting and receiving process.
Recommendations:
Avoid using acronyms in the abstract.
Define acronyms at first appearance in the text; see, for example, Sensitivity and fractional BandWidth (SBW), HDMS, etc.
Avoid using lumped references; see, e.g., [15-18], [19-22], [40-43], etc .; include a critical comment for each reference or at most 2 references.
In addition to mentioning the novelty of this study, include the main contributions at the end of the introduction.
Define the symbols used in relations (4) and (6).
Include a nomenclature.
Round 2
Reviewer 2 Report
The authors have addressed the comments and questions.